# Different Transcriptome Features of Peripheral Blood Mononuclear Cells in Non-Emphysematous Chronic Obstructive Pulmonary Disease

**DOI:** 10.3390/ijms25010066

**Published:** 2023-12-20

**Authors:** Takuro Imamoto, Takeshi Kawasaki, Hironori Sato, Koichiro Tatsumi, Daisuke Ishii, Keiichiro Yoshioka, Yoshinori Hasegawa, Osamu Ohara, Takuji Suzuki

**Affiliations:** 1Department of Respirology, Graduate School of Medicine, Chiba University, Chiba 260-8670, Japan; 2Department of Pediatrics, Graduate School of Medicine, Chiba University, Chiba 260-8670, Japan; 3Department of Applied Genomics, Kazusa DNA Research Institute, Chiba 292-0818, Japan; 4Synergy Institute for Futuristic Mucosal Vaccine Research and Development, Chiba University, Chiba 260-8670, Japan

**Keywords:** chronic obstructive pulmonary disease (COPD), peripheral blood mononuclear cells (PBMCs), RNA sequencing, emphysema, airway remodeling, hematopoietic cell lineage

## Abstract

Non-emphysematous chronic obstructive pulmonary disease (COPD), which is defined based on chest computed tomography findings, presented different transcriptome features of peripheral blood mononuclear cells (PBMCs) compared with emphysematous COPD. Enrichment analysis of transcriptomic data in COPD demonstrated that the “Hematopoietic cell lineage” pathway in Kyoto Encyclopedia of Genes and Genomes pathway analysis was highly upregulated, suggesting that cellular dynamic dysregulation in COPD lungs is affected by pathologically modified PBMCs. The differentially expressed genes (DEGs) upregulated in PBMCs reflected the disease state of non-emphysematous COPD. Upregulated DEGs such as *XCL1*, *PRKCZ*, *TMEM102*, *CD200R1*, and *AQP1* activate T lymphocytes and eosinophils. Upregulating keratan sulfate biosynthesis and metabolic processes is associated with protection against the destruction of the distal airways. *ITGA3* upregulation augments interactions with extracellular matrix proteins, and *COL6A1* augments the profibrotic mast cell phenotype during alveolar collagen VI deposition. Upregulating *HSPG2*, *PDGFRB*, and *PAK4* contributes to the thickening of the airway wall, and upregulating *SERPINF1* expression explains the better-preserved vascular bed. Therefore, gene expression and pathway analysis in PBMCs in patients with non-emphysematous COPD represented type 2 immune responses and airway remodeling features. Therefore, these patients have asthmatic potential despite no clinical signs of asthma, in contrast to those with emphysematous COPD.

## 1. Introduction

Chronic obstructive pulmonary disease (COPD) is an inflammatory disease characterized by airway and/or alveolar abnormalities that lead to persistent airflow limitations and respiratory symptoms. Approximately 25% of smokers develop COPD, suggesting that genetic factors and exposure to environmental stimuli are involved in COPD development [1,2]. The pathogenesis of COPD is complex, with heterogeneous clinical phenotypes. A classification of clinical phenotypes is based on chest computed tomography (CT) findings, and COPD is classified into two types: emphysematous and non-emphysematous. COPD endotypes can be determined by the different expressions of COPD-related genes that, in combination with individual susceptibility and environmental factors, may destroy the alveolar structure, cause airway wall remodeling, or cause vascular changes [3]. 

Peripheral blood mononuclear cells (PBMCs) are immune cells primarily classified into lymphocytes and monocytes. Lymphocytes can be categorized into T, B, and natural killer cells with various functions. Monocytes are CD14^+^ cells [4] and can be further categorized into three populations with different functions based on the combination of CD14 and CD16 expression levels. CD14^++^CD16^−^ monocytes are classified as classical, CD14^++^CD16^+^ monocytes as intermediate, and CD14^+^CD16^+^ monocytes as non-classical monocytes [4]. PBMCs are derived from bone marrow (BM) cells, and the information carried is reflected by germline and somatic mutations related to COPD pathogenesis. These gene directives in PBMCs may be essential in further developing COPD pathology.

Omics is widely used to study polygenic and phenotypically diverse diseases. These methods include genomics, epigenomics, transcriptomics, proteomics, and metabolomics. Many transcriptomic studies on COPD have been performed, focusing on alveolar macrophages and lung tissues [5,6,7]. Transcriptomics of bulk PBMCs, monocytes, and single PBMC cells has also been performed to elucidate the pathogenesis of many respiratory diseases, including COPD. Microarray and RNA sequencing are currently used for transcriptome analysis. However, RNA sequencing may provide more detailed information than microarrays for detecting comprehensive gene expression signatures.

Several studies on PBMCs in patients with COPD have suggested that genomic signatures are associated with COPD [5,6,8,9]. Although microarray studies have been performed in COPD [5,6,10], whether using RNA sequencing to determine the gene expression profiles in PBMCs of patients with COPD could lead to deeper insights into COPD pathogenesis remains unclear. Additionally, how PBMC gene expression reflects clinical phenotypic differences in emphysematous vs. non-emphysematous COPD types has not been examined. Therefore, in this study, we explored the gene expression characteristics of PBMCs from patients with non-emphysematous and emphysematous COPD.

## 2. Results

### 2.1. Differential Gene Expression and Pathway Analysis in PBMCs between Patients with COPD and Healthy Controls

This study included 26 patients with obstructive disorder-fixed COPD and 13 healthy controls (HCs). Four patients with a history of bronchial asthma were included when comparing COPD and HCs. The demographic characteristics of the participants are presented in Table 1. All patients with COPD had a history of smoking; six patients were current smokers, whereas the others with COPD had quit smoking long before sample collection. No significant differences in age or sex were observed between the patients with COPD and HCs. Details of the medications used by the patients with COPD are summarized in Table 2**.**

RNA sequencing libraries were prepared from mRNA isolated from the PBMCs of the participants. The RNA integrity values of all samples were >9, and 26,467 genes were obtained from the mRNA. Additional quality control methods were used to compare patients with COPD and HCs, and 12,189 genes were retained for further analysis. 

Principal component analysis (PCA) revealed that the two groups could be distinguished (Figure 1). We compared the differentially expressed genes (DEGs) in the PBMCs of patients with COPD and HCs. Figure 2 shows a volcano plot of the distribution of the log2-fold change and *p*-value for the 12,189 genes expressed in these samples. Of these, 119 genes were identified as DEGs (*p* < 0.05, fold change >2 or <0.5). Figure 3 shows a heat map of the 119 DEGs, with 36 downregulated and 83 upregulated genes in the PBMCs of patients with COPD compared with HC. The details of the DEGs are provided in Appendix A.

Enrichment analysis using the Enrichr online tool revealed that some gene ontologies and pathways were significantly enriched in the DEGs between patients with COPD and HCs. Gene ontology (GO) terms for upregulated and downregulated genes are listed in Table 3A. Kyoto Encyclopedia of Genes and Genomes (KEGG) pathway terms for upregulated and downregulated genes are listed in Table 3B. 

### 2.2. Differential Gene Expression and Pathway Analysis in PBMCs between Non-Emphysematous and Emphysematous COPD

In the following analysis, we first excluded patients with a history of bronchial asthma (*n* = 4) to get closer to the omics characteristics of non-emphysematous COPD. Subsequently, we classified patients with COPD into two groups: non-emphysematous (*n* = 14) and emphysematous (*n* = 8), according to the extent of emphysematous changes in CT images. Each patient with COPD was classified into non-emphysematous or emphysematous COPD according to Goddard’s criteria [11]. The demographic characteristics of the participants are presented in Table 4. FEV_1_/FVC and %FEV_1_ between the two groups differed significantly (*p* < 0.05), whereas no significant differences were observed regarding sex, smoking index, or body mass index.

After processing the RNA sequencing libraries, 12,414 DEGs were observed in patients with emphysematous and non-emphysematous COPD. The samples were divided into two groups (Figure 4). The volcano plot in Figure 5 shows the distribution of log2-fold changes and *p-*value for the 12,414 genes expressed in these samples. Of these, 183 genes were identified as DEGs (*p* < 0.05, fold change >2 or <0.5). Figure 6 shows a heat map of 183 DEGs, with 121 upregulated and 62 downregulated genes in the PBMCs of patients with non-emphysematous COPD compared with those with emphysematous COPD. The details of the DEGs are provided in Appendix A.

Enrichment analysis using the Enrichr online tool revealed significant enrichment in some DEG ontologies and pathways between patients with emphysematous and non-emphysematous COPD (Table 5). GO terms with downregulated genes included “Regulation of vasoconstriction” and “Peptidyl-proline modification” (Table 5A). KEGG pathway terms with downregulated genes included “Vascular smooth muscle contraction” (Table 5B).

## 3. Discussion

COPD is not a single disease but a cluster of several diseases sharing the common clinical feature of irreversible airflow limitation. Airflow limitation in COPD is primarily caused by inflammation-induced structural changes in the small airways (obstructive bronchiolitis) or destruction of the lung parenchyma caused by proteases and antiproteases (emphysema) activity imbalance. Inflammation-induced structural changes cause fibrosis or luminal plugs in the small airways. In contrast, airway obstruction in emphysema is induced by the loss of alveolar attachments to the small airways, reducing lung elastic recoil. Airway and lung parenchymal immune cells are hallmarks of chronic inflammatory lung diseases, such as COPD and asthma, and understanding the mechanisms that promote increased lung inflammatory processes is crucial for effective pharmacotherapeutic development. Expansion of hematopoietic compartments in the BM promotes the differentiation and trafficking of mature inflammatory cells to the airways. Hematopoietic progenitor cells are derived from the BM and migrate to the lungs, where in situ differentiation, according to the tissue microenvironment, provides a source of pro-inflammatory cells. In addition, hematopoietic progenitor cells in the airways respond to locally derived damage to produce humoral factors, acting as effector pro-inflammatory cells that potentiate lung inflammation [12]. Therefore, the expression of the PBMC gene signature partly reflects information on hematopoietic progenitor cells. 

### 3.1. COPD vs. HCs

In this study, transcriptome analysis using RNA sequencing indicated that the gene expression profiles of bulk PBMCs differed between patients with COPD and HCs, revealing 119 differentially expressed genes. GO and KEGG pathway enrichment analyses demonstrated the enrichment of various biological processes and pathways related to airway, vascular, and tissue remodeling (Table 3A,B). Regarding the upregulated DEGs in COPD, *CD109*, B cell linker (*BLNK*), interleukin-1 receptor type 1 (*IL1R1*), *CD1A*, and carboxypeptidase A3 (*CPA3*) are related to T-helper cell type 2 (Th2) inflammation, but not to emphysema formation. Tissue factor pathway inhibitor (*TFPI*), protein S1 (*PROS1*), and membrane metallo-endopeptidase (*MME*) likely act against pulmonary vascular remodeling. Colony-stimulating factor 1 (*CSF-1*) and *TFPI* are involved in BM-derived progenitor cell function. Sialidase NEU3*3* (*NEU*) and G protein-coupled estrogen receptor 1 (*GPER1*) are related to cell proliferation. DNA cross-link repair 1A (*DCLRE1A*) and Nei-Like DNA Glycosylase 3 (*NEIL3*) are DNA repair genes. *ELANE* is an elastase associated with emphysema. Our results revealed that genes related to the immune system and inflammatory responses were upregulated in the PBMCs of patients with COPD [9].

*ELANE* was upregulated among the GO term “Acute inflammatory response” (Table 3A). Elastases are a subfamily of serine proteases that hydrolyze many proteins and elastin. Humans possess six elastase genes that encode structurally similar proteins, and activated proteases hydrolyze proteins within specialized neutrophil lysosomes and proteins in the extracellular matrix (ECM). This enzyme plays a role in emphysema formation, such as in COPD, through collagen-IV and elastin proteolysis [13]. Neutrophil elastase inhibits the expression of elastic fiber assembly components and downregulates TGF-β signaling in mice [14]. 

The GO term “Positive regulation of EGFR signaling” included *NEU3* and *GPER1* in the COPD group. Sialidase NEU3, a key glycosidase for ganglioside degradation, is upregulated in various human cancers, increasing cell invasion, motility, and survival of cancer cells, possibly through activating epidermal growth factor (EGF) signaling [15]. However, the role of NEU3 in COPD progression is unclear. Estrogen functions via GPER/Gαi signaling to modulate the EGFR/ERK and HIF-1α/TGF-β1 signaling to increase prostatic stromal cell proliferation and fibrosis [16]. *NEU3* and *GPER1* are associated with cell proliferation; however, the roles of these genes in COPD have not yet been determined.

The upregulated GO term “Negative regulation of wound healing” in COPD includes *CD109* and *TFPI*. Dendritic cells (DCs) are essential for asthma development because they contain allergens that cause Th2 inflammation. CD109, a glycosylphosphatidylinositol-anchored glycoprotein in DCs**,** is involved in airway hyperreactivity and allergic inflammation. CD109-deficient mice had reduced AHR and eosinophilic inflammation and lower Th2 cytokine expression than wild-type mice [17]. Thus, CD109 upregulation is possibly associated with Th2 inflammation in COPD airways.

A clinical feature of COPD is pulmonary vascular endothelial dysfunction, chronic obstructive bronchiolitis, and emphysema. Functional changes in pulmonary endothelial cells in COPD include anti-coagulant disturbances involving changes in *TFPI* expression. *TFPI* is a single-chain polypeptide that reversibly inhibits factor Xa. Endothelial disturbance could be reversible in the early stages of COPD, as endothelial repair/regeneration occurs with upregulating vascular endothelial growth factors and increased BM-derived progenitor cells [18]. The upregulation of *TFPI* indicates the involvement of pulmonary vascular endothelial impairment during COPD development.

The upregulation of *PROS1* and *TFPI* are included in the GO term “Complement and coagulation cascades”. Patients with COPD have an increased risk of cardiovascular disease and venous thromboembolism because stable COPD may exhibit increased coagulation factor levels and decreased coagulation inhibitor levels, resulting in a prothrombotic state [19]. Protein S, also known as PROS, is encoded by *PROS1* [20], a vitamin-K-dependent plasma glycoprotein synthesized in the liver and a coenzyme of complement protein C4b-binding protein, which exhibits coagulation inhibitory effects. *PROS1* and *TFPI* upregulation could act against the prothrombotic state of COPD.

*DCLRE1A* and *NEIL3*, included in “Interstrand cross-link repair”, were upregulated in patients with COPD; however, the role of these genes in COPD pathogenesis is unclear. Inadequate DNA repair has been implicated in the pathogenesis of COPD; however, DNA damage is repaired by an integrated network of cellular signaling pathways and the DNA damage response [21]. *DCLRE1A* participates in repairing cross-linked DNA, whereas *NEIL3* is involved in the base excision DNA repair pathway. 

The upregulation of *BLNK* and *IL1R1* are included in the GO term “NF-kappa B signaling pathway”. Adaptive humoral immune responses in the airways are mediated by B and plasma cells, which express highly evolved and specific receptors and produce immunoglobulins. The BLNK protein is expressed in B cells and macrophages and plays a substantial role in B cell receptor signaling. Antigen exposure in the upper or lower airways can drive B-lineage cell expansion in airway mucosal tissue, forming inducible lymphoid follicles or aggregates mediating local immunity or disease. [22]. Interleukin (IL)-1β is a typical innate immune cytokine involved in the initiation and persistence of inflammation. The signaling receptor for IL-1β is *IL1R1*, and the decoy receptor is *IL1R2.* Persistent IL-1 signaling activation through IL-1R1 influences airway inflammation in patients with COPD and asthma [23]. Therefore, the upregulation of *BLNK* and *IL1R1* may be associated with airway inflammation rather than emphysema formation.

#### Hematopoietic Cell Lineage

The term “Hematopoietic cell lineage” was upregulated in the KEGG pathway analysis (Table 3B). *CSF1*, *MME*, *CD1A*, *IL1R1*, *IL1R2*, and *HLA-DRB5* were included in the current study. *CD1A*, which encodes a CD1 transmembrane protein family member, is expressed in antigen-presenting cells (APCs). CD1 proteins primarily mediate the presentation of lipid and glycolipid antigens of self- or microbial origin to T cells. Transcript *CD1A* levels are upregulated in the lung parenchyma of smokers [24]. The cigarette smoke-mediated loss of C1q, a complement protein 1 complex (C1) component, is key in reducing peripheral tolerance. C1q potentiates the function of APCs to differentiate CD4^+^ T cells into Tregs while inhibiting Th17 cell development and proliferation [25]. CD1 proteins seem to act at APCs, not to differentiate CD4^+^ T cells to regulatory T cells (Tregs) in cigarette smokers.

*CSF-1*, which includes macrophage colony-stimulating factor (M-CSF), stimulates hematopoietic progenitor cells to differentiate into several monocyte/macrophage lineage cells. However, M-CSF is minimally involved in the development and cellular responses of alveolar macrophages (AMs) [26]. Thus, unlike several monocyte/macrophage lineage subsets, AMs likely rely on factors other than M-CSF for their development. The granulocyte-macrophage colony-stimulating factor is critical for AM development. However, the role of *CSF-1* upregulation in patients with COPD remains unknown.

The *MME* gene encodes neprilysin, membrane metallo-endopeptidase, neutral endopeptidase, and CD10. CD10 expressed-hematopoietic progenitor cells are common progenitor cells of the lymphoid lineage. These cells can differentiate into T, natural killer (NK), and B cells. Neprilysin activity and expression are substantially decreased in human lungs with COPD and in isolated human pulmonary arterial smooth muscle cells exposed to cigarette smoke extract and hypoxia via mechanisms that include oxidative reactions and protein degradation [27]. Therefore, if *MME*, including neprilysin, is upregulated in patients with COPD, it may prevent vascular remodeling and pulmonary hypertension complicated by COPD.

Since COPD is not a disease but a collection of various phenotypes, association studies regarding the *HLA-DRB5* locus and its alleles have attracted attention regarding COPD susceptibility. HLA-DRB5 is highly expressed in the PBMCs of patients with systemic sclerosis (SSc)-related interstitial lung diseases, and the HLA-DRB5*01:05 allele was possibly a risk factor for ILD in patients with SSc [28].

The term “Protein digestion and absorption” was upregulated in the KEGG pathway analysis (Table 3B). *CPA3*, collagen type XXIV alpha 1 chain (*COL24A1*), and *MME* were included in this study. COPD is a highly heterogeneous and complex condition characterized by chronic airway inflammation and remodeling. A subset of patients with COPD has increased eosinophil infiltration into the airways. Mast cell (MC) gene expression of CPA3 was increased in patients with eosinophil-high compared to those with eosinophil-low COPD. Eosinophilic Th2 inflammation in patients with COPD involves changes in mast cell characteristics. *CPA3* upregulation could be reflected by airway inflammation and not by emphysema formation. *MME* functions at the pulmonary vasculature, as discussed in “Hematopoietic cell lineage”. However, the role of *COL24A1* in COPD has not yet been elucidated.

### 3.2. Emphysematous vs. Non-Emphysematous COPD

We first excluded patients with a history of bronchial asthma in the subsequent comparison between patients with non-emphysematous and emphysematous COPD (Table 5A,B). The patients with non-emphysematous COPD had no history of asthma or large bronchodilator reversibility, and they had no exacerbations or hospital admissions during the past year. Nevertheless, in non-emphysematous COPD, various biological processes and pathways related to the type 2 immune response and airway remodeling, including a thickened distal airway, were enriched compared with emphysematous COPD (Table 5A,B).

Patients with non-emphysematous COPD did not clinically belong to the definition of asthma—COPD overlap syndrome (ACOS) (Table 4). ACOS captures a subset of patients with airway diseases with features of asthma and COPD. Although the definitions of ACOS vary, ACOS generally encompasses persistent airflow limitation in patients older than 40 years with a history of asthma or large bronchodilator reversibility. ACOS affects approximately a quarter of patients with COPD and almost one-third of patients who previously had asthma. Compared with their counterparts with asthma or COPD alone, patients with ACOS have significantly worse respiratory symptoms, poorer quality of life, and an increased risk of exacerbations and hospital admissions. In the present study, gene expression and pathway analysis in PBMCs in patients with non-emphysematous COPD represented some of the features of type 2 immune response and airway remodeling, suggesting that these patients have asthmatic potential despite no clinical signs of asthma, compared to those with emphysematous COPD.

#### 3.2.1. GO Terms Related to T Lymphocytes

In patients with non-emphysematous COPD, increased gene expressions were observed in many GO terms related to T lymphocytes, such as “Positive regulation of T-helper 2 cell cytokine production”, “Regulation of T-helper 2 Cell cytokine production”, “Positive regulation of Type 2 immune Response”, “Regulation of T cell cytokine production”, “Positive Regulation of T cell cytokine production”, “Regulation of T cell migration”, and “Positive regulation of T cell migration”. GO analysis suggests that the type 2 immune response is upregulated in non-emphysematous COPD.

X-C motif chemokine ligand 1 (*XCL1*, lymphotactin) and protein kinase C zeta (*PRKCZ*) were included in four GO terms (Table 5). *XCL1* is expressed in many lymphocyte subsets, such as T and NK cells [29], and recruits lymphocytes to areas of allergic inflammation [30,31]. Additionally, XCL1 contributes to the functional maintenance of Tregs [32]. In an in vitro human blood eosinophil migration study using ECM components, T-cell protein kinase C ζ (PKC-zeta) was largely involved in eosinophil migration; however, its specific targets remain undefined [33]. Therefore, *XCL1* and *PRKCZ* upregulation in patients with non-emphysematous COPD may contribute to the recruitment of lymphocytes and eosinophil migration, suggesting that stronger activation of lymphocytes and eosinophils is associated more with non-emphysematous COPD than with emphysema-dominant COPD.

Temperature triggers the development and exacerbation of asthma. Cold and hot air provoke inflammation and bronchial remodeling by expressing ankyrin-like and mela-statin receptors. The reception of physical and chemical environmental stimuli and body temperature regulation is mediated by thermo-sensory channels, members of a subfamily of transient receptor potential (TRP) ion channels. *TRPM* polymorphisms have been established in cold airway hyperreactivity, and *TRPM* knockdown attenuates cold-induced inflammation, reduces Th1/Th2 imbalances, and positively affects airway remodeling. Genes encoding mela-statin TRP channels, such as *TRPM4*, are involved in developing some asthma phenotypes and exacerbating asthma [34].

Transmembrane protein 102 (*TMEM102*), which was upregulated in the GO term “Positive regulation of T cell migration” in patients with non-emphysematous COPD, is involved in regulating mitochondrial outer membrane permeabilization in the apoptotic signaling pathway, responding to cytokines, and signal transduction. *TMEM102* acts upstream of the positive regulation of T-cell migration and adhesion. Naive CD4^+^ T cells differentiate into functionally diverse Th cell subsets, and Th2 cells play a pathogenic role in asthma. Th2 cells in the airways are enriched in the transcription of CD200 receptor 1 (*CD200R1*), although the role *CD200R1* plays on Th2 cells remains unclear. However, a single-cell RNA sequencing study of Th-cell responses to house dust mites suggested an autoregulatory function of the CD200R1-ligand pair in early Th2 cell differentiation [35]. Therefore, *TMEM102* and *CD200R1* upregulation in “Regulation of T cell migration” may contribute to Th cell activation in the non-emphysematous COPD subgroup.

#### 3.2.2. Keratan Sulfate Biosynthesis and Metabolic Process

Reducing the levels of airway glycan ligands for Siglec-F diminishes the natural pro-apoptotic pathway for controlling airway eosinophilia and is associated with reduced ST3Gal-III function in the lungs, resulting in exaggerated eosinophilic airway inflammation [36]. Upregulated ST3 beta-galactoside alpha-2,3-sialyltransferase 3 (*ST3GAL3*) may induce airway ligands for Siglec-F, resulting in preserving the ability of the natural pro-apoptotic pathway to control airway eosinophilia.

The keratan sulfate (KS) biosynthesis and metabolic process-related genes *ST3GAL3* and *B3GNT7* are upregulated in patients with non-emphysematous COPD; however, the role of *B3GNT7* has not been examined. KS, one of the major glycosaminoglycans produced in the small airways, was decreased in the lungs of cigarette smoke-exposed mice. Its biological function is associated with KS proteoglycans, and inflammation-induced KS-linked protein degradation may be partially responsible for reduced KS [37]. The upregulation of KS biosynthesis and metabolic processes in the non-emphysematous COPD subgroup suggests its ability to protect against the destruction of distal airways.

#### 3.2.3. ECM-Receptor Interaction

KEGG pathway analysis showed upregulation of the term “ECM-receptor interaction”, which includes integrin subunit alpha 3 (*ITGA3*), collagen type VI α1 chain (*COL6A1*), and heparan sulfate proteoglycan 2 (*HSPG2*) in the non-emphysema subgroup.

Although they have fixed obstructive impairment, patients with non-emphysematous COPD show fewer emphysematous lesions on high resolution computed tomography, suggesting that non-emphysematous phenotypes exhibit prominent distal airway lesions. Although they do not show any asthmatic symptoms, they could have a common pathobiology with refractory asthma since non-emphysematous COPD and refractory asthma have distal airway remodeling. During the airway epithelium repair after asthma injury, cell adhesion and migration, which depend on the interactions between ECM proteins and appropriate integrins, are required. Integrins expressed by airway epithelial cells mediate wound closure through different constitutive ECM proteins. α-Integrin subunits (α2-, α3-, and α6-integrin) alone do not mediate epithelial cell migration grown on laminin-1 or -2 matrix in response to EGF in a cultured human airway epithelial cell line [38]. Therefore, *ITGA3* upregulation in non-emphysematous COPD may augment interactions between ECM proteins and integrin subunit alpha 3.

Refractory asthma is associated with increased collagen deposition in the conducting airways and the alveolar parenchyma. MCs in asthma affect collagen synthesis. Patients with uncontrolled atopic asthma have an altered profibrotic MC phenotype in the alveolar parenchyma associated with alveolar collagen VI [39]. Thus, *COL6A1* upregulation in patients with non-emphysematous COPD could augment the profibrotic mast cell phenotype regarding alveolar collagen VI deposition.

In non-emphysematous COPD, airway remodeling in the distal airways is involved in pathophysiology. Fibroblasts in the central and distal airways differ in proteoglycan production and proliferation, even in patients with mild asthma [40]. Thus, *HSPG2* upregulation in non-emphysematous COPD may play a role in distal airway remodeling.

#### 3.2.4. Focal Adhesion

Platelet-derived growth factor receptor beta (*PDGFRB*), *ITGA3*, *COL6A1*, and p21 (RAC1)-activated kinase 4 (*PAK4*) were included in the GO term “Focal adhesion”, in which DEGs were upregulated in patients with non-emphysematous COPD. Platelet-derived growth factor (PDGF) is a cytokine that thickens the airway walls by increasing the smooth muscle and connective tissue in COPD and asthma. PDGF is immunolocalized to tissue macrophages, and PDGF receptor-beta is occasionally expressed in the bronchial epithelium. PDGFB and PDGFR-beta mRNA in lung tissue appear more frequently in patients with asthma than with COPD; however, PDGF and its receptor do not correlate closely with structural changes in diseased airways [41]. *PDGFRB* upregulation in non-emphysematous COPD may contribute to airway pathology.

The establishment of asthma pathobiology may be driven by Th lymphocytes, with eosinophils as major effector cells. Recruitment of inflammatory cells from the blood to the airways is mediated by adhesive molecules, such as selectins and integrins. Cell surface receptors and ECM ligands mediate cell migration to the inflammatory site through blood vessels. Integrins, a family of heterodimeric glycoproteins comprising α and β subunits, play an essential role in this process. In asthma development, the most important in cell trafficking are integrins containing α4 and β2 subunits [42]. α1β1 and α2β1 integrins are distributed on leukocytes, which may be involved in asthma [43]. Leukocyte infiltration may be related to non-emphysematous COPD pathogenesis; however, no previous reports have examined *ITGA3* expression in COPD.

Increased collagen deposition in the conducting airways, alveolar parenchyma, and MCs affect collagen synthesis in asthma. The proportion of TGF-β positive phenotype of MCs correlated positively to an increased immunoreactivity of alveolar collagen VI but not with collagen I and III. Furthermore, collagen VI was increased in the alveolar parenchyma of patients with uncontrolled asthma compared with those with controlled asthma, suggesting that uncontrolled atopic asthma has altered profibrotic MCs in the alveolar parenchyma associated with alveolar collagen VI [39]. *COL6A1* upregulation could be associated with increased collagen deposition in the alveolar parenchyma, possibly related to non-emphysematous COPD development.

The Rho GTPase superfamily can be categorized into three subfamilies: Rho, Rac, and Cdc42, which mediate smooth muscle contractions. Rac1-related pathways are involved in bronchial smooth muscle contractions, and Rac1 inhibition prevents bronchoconstriction and airway hyper-responsiveness [44]. An increase in Rac1-mediated signaling is involved in the augmented contractions of bronchial smooth muscles in mice with antigen-induced airway hyper-responsiveness [45]. The upregulation of PAK4 may be involved in airway smooth muscle cell contraction in patients with non-emphysematous COPD.

#### 3.2.5. Hedgehog Signaling Pathway

“Hedgehog signaling pathway” was upregulated in KEGG pathway analysis in patients with non-emphysematous COPD. Hedgehog signal transduction abnormalities lead to asthmatic airway remodeling, characterized by cellular matrix modification, collagen deposition, epithelial cell proliferation, epithelial-mesenchymal transition, and fibroblast activation [46]. Hedgehog signaling also targets a function in tissue repair and immune-related disorders of asthmatic airways [47]. Considering the common pathobiology between non-emphysematous COPD and asthma, this signaling pathway could play a role in forming airway lesions. However, the specific roles of G protein-coupled receptor 161*1* (*GPR16*) and EF-hand calcium binding domain 7*7* (*EFCAB*), observed in this study, have not been identified.

#### 3.2.6. Angiogenesis Regulation

COPD pathogenesis has been partly explained by the relationship between neutrophil serine protease, neutrophil elastase, and their endogenous inhibitor, alpha-1-antitrypsin [48]. Serpin family F member 1*1* (*SERPINF*) is better known as pigment epithelium-derived factor (PEDF). An imbalance between the pro-angiogenic activities of vascular endothelial growth factor and the anti-angiogenic activities of PEDF could partly explain the reduced vascular density in COPD. *SERPINF1* upregulation could explain the better-preserved vascular bed in the non-emphysematous COPD subgroup.

Isthmin 1 (*ISM1*) protects lung homeostasis via apoptosis of AMs that harbor high levels of its receptor, cell surface GRP78. ISM1 is a lung-resident anti-inflammatory protein that selectively triggers AM apoptosis. ISM1 knockout mice (Ism1-/-) show increased functional heterogeneity in AMs, with enduring lung inflammation and progressive emphysema, similar phenotypes to those in human COPD [49].

Aquaporins (AQPs) are water-permeable channels responsible for cellular fluid transport across the cells. AQP-1 plays a role in fluid flux and eosinophil movement, whereas AQP-5 is involved in pulmonary hyper-responsiveness and airway inflammation in asthma [50]. Regarding the genes regulating angiogenesis, *SERPINF1* and *ISM1* upregulation protect against emphysema formation, whereas *AQP1* upregulation promotes eosinophil infiltration.

#### 3.2.7. Interim Summary Regarding the Genetic Profile of Non-Emphysematous COPD

*XCL1* and *PRKCZ* may play roles in lymphocyte recruitment and eosinophil migration. *TRPM4* is associated with cold airway hyperreactivity, *TMEM102* and *CD200R1* are involved in Th cell activation, and *ST3GAL3* and *B3GNT7* are associated with protecting the distal airways against destruction. *ITGA3* augmented interactions between ECM proteins and ITGA3. *COL6A1* may augment the profibrotic MC phenotype during alveolar collagen VI deposition. *HSPG2* upregulation may play a role in distal airway remodeling. *PDGFRB* can thicken the airway walls by increasing smooth muscle and connective tissue. *ITGA3* expression is associated with leukocyte infiltration. *COL6A1* may be associated with increased collagen deposition in the alveolar parenchyma. PAK4 is involved in smooth muscle cell contraction in the airway. The Hedgehog signaling pathway (*GPR161* and *EFCAB7*) leads to asthmatic airway remodeling. *SERPINF1* and *ISM1* protect against emphysema formation, whereas *AQP1* promotes eosinophilic infiltration. Adrenomedullin and myosin light chain kinase family member 4 (*MYLK4*) act against emphysema.

### 3.3. Study Limitations

Although the q-value of the DEGs (Appendix A) should be noted and the DEGs revealed in this study require independent confirmation, our findings provide an aspect of the pathobiological concepts of COPD and non-emphysematous COPD to be further explored. COPD is a complex condition, and even in a large and well-controlled COPD cohort (ECLIPSE), the clinical manifestations, including the amount of emphysema observed using CT, are highly variable. Furthermore, the degree of airflow limitation, a definition of COPD, does not capture the heterogeneity of the disease [51]. Nevertheless, analyzing transcriptomic data in PBMC seems to be reflected by an aspect of clinical characteristics in non-emphysematous COPD. Further investigations from the point of view of COPD and asthma and further exploration of their combination would validate our findings. Additionally, our results suggest using molecular profiling to stratify patients with COPD. To implement this study with the analysis of the predictive/differentiating potential of the identified gene expression signatures would be valuable.

Although characterization of emphysema vs. non-emphysema phenotypes could lead to more targeted therapies for COPD, phenotyped COPDGene cohort cluster analysis of hundreds of COPD-associated features showed heterogeneous effects of genetic variants on COPD-related phenotypes, including CT measurements of airway abnormalities and emphysema [52,53,54,55].

The present study demonstrated that “Hematopoietic cell lineage” was the most significantly upregulated in patients with COPD. Although our study was conducted to clarify the characteristics of non-emphysematous COPD, selection bias of patients may have occurred. Chest CT examinations are performed in most COPD cases in clinical practice in Japan, and the non-emphysematous type of COPD accounts for approximately 10% of patients. Because aging is an essential factor in COPD establishment, we focused on matching the ages of both groups. The COPDGene study has been one of the largest studies to investigate the underlying genetic factors of COPD, and clonal hematopoiesis of indeterminate potential was associated with odds ratios of 1.6 for GOLD 2-4 and 2.2 for GOLD 3-4 COPD compared with those for noncarriers, in a multivariable model that included age, smoking, and sex. Patients included in COPDGene undergo comprehensive phenotyping, including chest CT scans and exclusions for other lung diseases [56]. Our small cohort study revealed that RNA sequencing analysis to characterize gene expression in PBMCs would be useful for identifying the general direction of gene expression; however, racial differences may exist in gene characteristics. 

Although the DEGs revealed in this study require independent confirmation, our findings provide an aspect of the pathobiological concepts of non-emphysematous COPD to be further explored. Therefore, further investigations focusing on COPD and asthma and exploring their combination would validate our findings.

Our results are similar to those of a previous study that used online databases from microarrays and revealed the upregulation of genes related to the immune system and inflammatory responses in the PBMCs of patients with COPD [9]. However, the results of this study revealed that recent advances in genetic analysis methods could provide a comprehensive understanding of COPD development.

## 4. Materials and Methods

### 4.1. Participants

This study was approved by the Human Ethics Committee of Chiba University (protocol no. 2083). PBMCs were collected from patients diagnosed with COPD or HCs between August 2020 and February 2023 at Chiba University Hospital. Patients with COPD met the Global Initiative for Chronic Obstructive Lung Disease criteria, whereas HCs were defined as those without apparent respiratory diseases regardless of their smoking history. 

### 4.2. Isolation of PBMCs

Peripheral blood was collected using a BD Vacutainer CPT Cell Preparation Tube with sodium citrate according to the manufacturer’s protocol (#362760) (Becton, Dickinson and Company, East Rutherford, NJ, USA). Tubes containing blood were centrifuged at 1500 rpm at room temperature for 20 min. After centrifugation, the plasma was removed from the uppermost layer. The PBMC layer was transferred to 15-mL conical tubes, washed with phosphate-buffered saline twice, followed by adding Isogen (Nippongene, Tokyo, Japan), and stored at −80 °C.

### 4.3. Total RNA Extraction, mRNA Library Preparation, and 3’ RNA Sequencing

Total RNA was extracted from ~2.0 × 10^6^ PBMCs, and each sample was transferred to 1.0 ml of Isogen reagent (Life Technologies, Carlsbad, CA, USA). The solution was then vortexed vigorously and incubated at room temperature for 5 min. The solution was centrifuged after adding chloroform, and the aqueous phase was carefully transferred to a new tube, after which 10 mg of glycogen (Life Technologies) was added as a co-precipitant. RNA was precipitated by adding 600 μL of isopropyl alcohol. The RNA pellet was washed once with 75% ethanol and dissolved in 10 μL of RNase-free water. The concentration and quality of the RNA were verified using a Qubit fluorometer (Life Technologies) and an Agilent 2100 Bioanalyzer (Agilent Technologies, Santa Clara, CA, USA), respectively. Purified total RNA (200 ng) was used for RNA library preparation, according to the instructions of the Quant Seq 3′ mRNA-seq library preparation kit FWD for Illumina (Lexogen, Vienna, Austria). RNA libraries were sequenced on an Illumina NextSeq 500 system with 75-nt-long reads. The FASTQ files were prepared with reads using bcl2fastq ver2.20 (Illumina). The quality of the FASTQ sequence data was assessed using FastQC v0.11.9 (Illumina). After removing adapter sequences from the raw reads, the trimmed reads were aligned using STAR v2.7.6a to the GRCh38 human reference genome. Reads per million (RPM) values were calculated using samtools v1.11 and htseq count v0.12.4.

### 4.4. 3’ RNA Sequencing Data Analysis

The RPM data were log2-transformed and filtered to ensure that at least one group contained at least 70% valid values for each gene. The remaining missing values were imputed using random numbers drawn from a normal distribution (width = 0.3; downshift = 1.8). The unpaired Student’s *t*-test was used to compare the two groups. Statistical significance was defined as a two-sided *p*-value < 0.05. The false discovery rate was also calculated as a q-value and taken into consideration to interpret the *p*-values. RNA sequencing RPM data were analyzed using Qlucore Omics Exploration software ver. 3.9.9 (Qlucore AB, Lund, Sweden). The fold changes between each group (>2.0 upregulated or <0.5 downregulated) with *p-*values of <0.05 were set to detect DEGs. GO and Kyoto KEGG pathway analyses were performed using the Enrichr online tool (http://amp.pharm.mssm.edu/Enrichr/) for enrichment analysis of DEGs (accessed on 1 August 2023). The gene set databases used in this study were “GO_Biological_Process_2021” (Terms: 6036; gene coverage: 14,937) and “KEGG_2021_Human” (Terms: 320; gene coverage: 8078). The GO terms and KEGG pathways were considered statistically significant at *p* < 0.05.

### 4.5. Statistical Analyses

The age characteristics of the samples are expressed as the means ± standard deviations. Student’s t-test was used for age comparisons. We compared the sexual characteristics of the samples using Fisher’s exact test. Statistical significance was set at *p* < 0.05.

## 5. Conclusions

The current RNA sequencing study suggests that bulk gene expression patterns in PBMCs differ between HCs and patients with COPD and between patients with emphysematous and non-emphysematous COPD. Changes in the gene expression patterns of PBMCs reflect the presence of COPD and explain some aspects of non-emphysematous COPD. These novel findings are crucial for strengthening our understanding of the complexities of the etiology and pathobiology of COPD and may lead to identifying potential therapeutic targets for COPD.

## Figures and Tables

**Figure 1 ijms-25-00066-f001:**
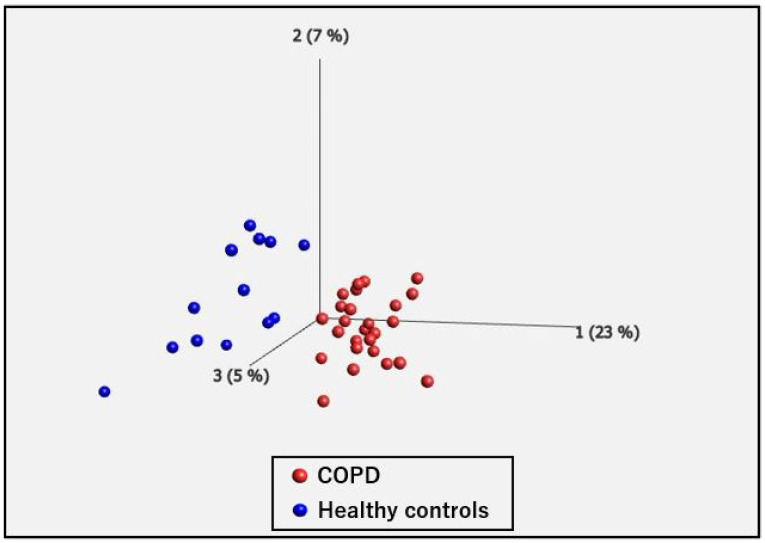
Principal component analysis (PCA). PCA shows two well-differentiated groups of chronic obstructive pulmonary disease (COPD) and healthy controls.

**Figure 2 ijms-25-00066-f002:**
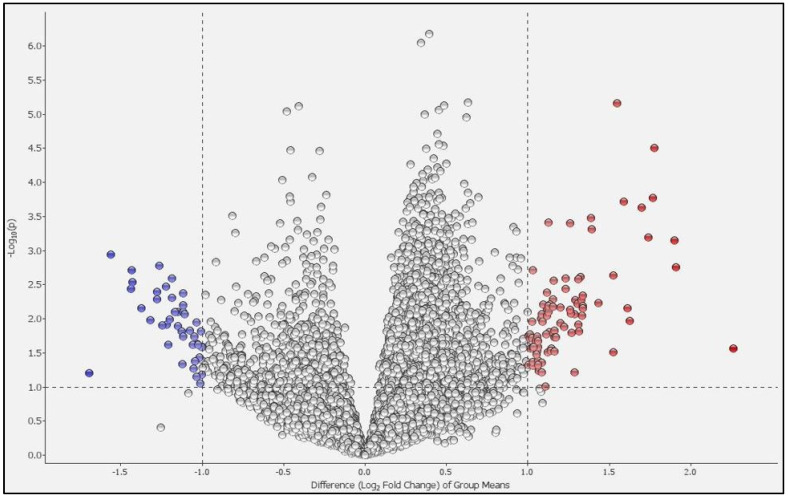
Volcano plot of differentially expressed genes (DEGs). The distribution of log_2_-fold change and *p*-values for the 12,189 genes are shown. Dots highlighted in color are the 119 DEGs between patients with COPD and healthy controls. Red dots represent high expression genes, whilst blue dots represent low expression genes in patients with COPD compared to healthy controls.

**Figure 3 ijms-25-00066-f003:**
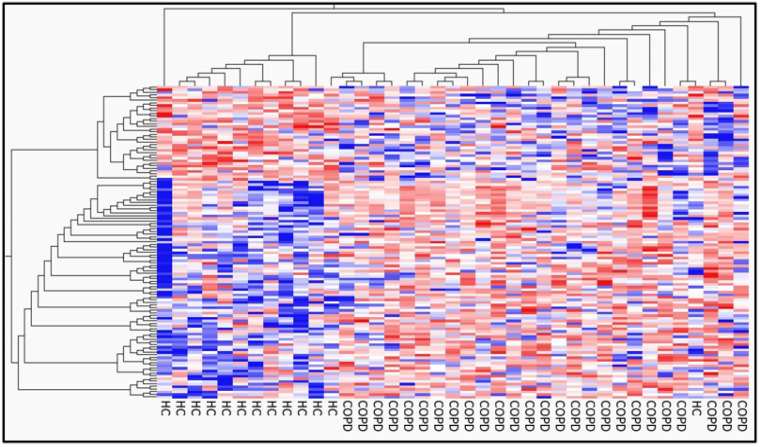
Heatmap of DEGs. The heatmap shows the DEGs between patients with COPD and healthy controls (HCs). Red bars represent high expression, and blue bars represent low expression.

**Figure 4 ijms-25-00066-f004:**
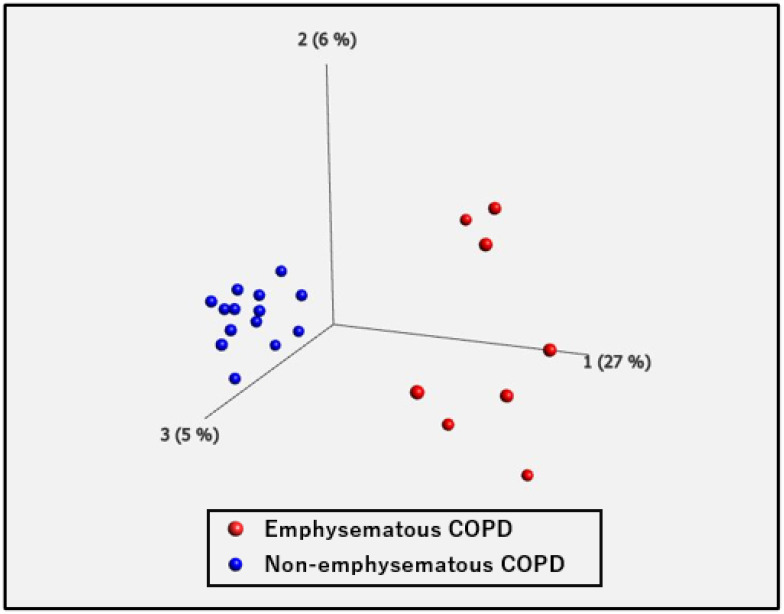
Principal component analysis (PCA). PCA shows two well-differentiated groups of patients with emphysematous and non-emphysematous COPD.

**Figure 5 ijms-25-00066-f005:**
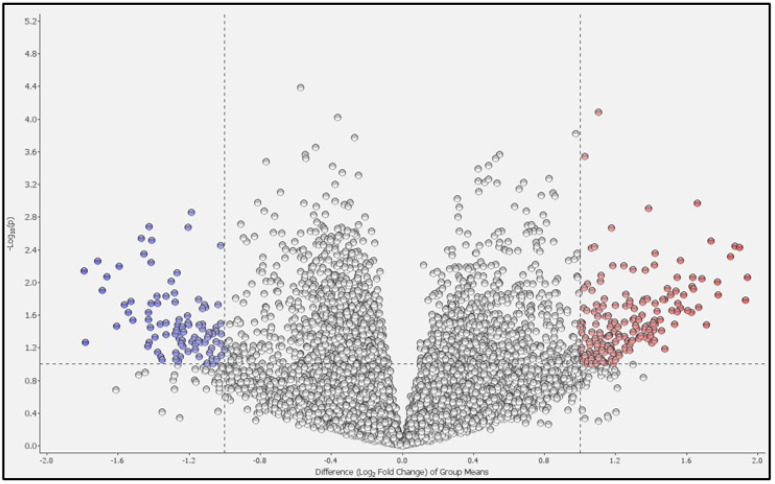
Volcano plot of differentially expressed genes (DEGs). The distribution of log_2_-fold change and *p*-value for the 12,414 genes are shown. Dots highlighted in color are 183 DEGs between non-emphysematous and emphysematous COPD. Red dots represent high expression genes, whilst blue dots represent low expression genes in patients with non-emphysematous COPD compared to emphysematous.

**Figure 6 ijms-25-00066-f006:**
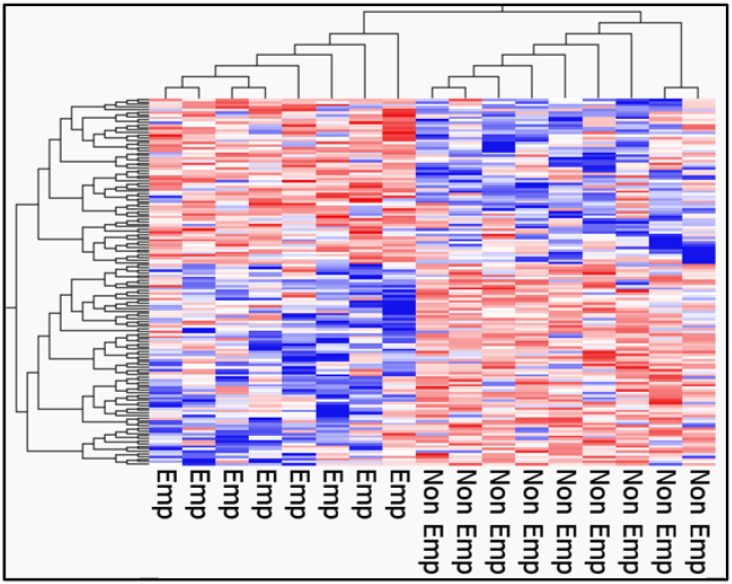
Heatmap of DEGs. The heatmap shows the DEGs between emphysematous (Emp) and non-emphysematous (Non Emp) COPD. Red bars represent high expression, and blue bars represent low expression.

**Table 1 ijms-25-00066-t001:** Characteristics of the patients with chronic obstructive pulmonary disease (COPD) and healthy controls.

	COPD	Healthy Controls	*p*-Value
Number of participants	26	13	N/A
Sex (male/female)	22/4	10/3	*p* = 0.51
Age	68.0 ± 0.3	66.5 ± 0.6	*p* = 0.54
BMI (kg/m^2^)	23.5 ± 3.7	N/A	N/A
Current smokers, number (%)	6 (23.1)	1 (7.7)	*p* = 0.39

Data are shown as the mean ± SD, median (interquartile range), or number (%). BMI, body mass index. N/A, not available.

**Table 2 ijms-25-00066-t002:** Medications used by patients with COPD.

Treatment	Number (%)
ICS	2 (7.7)
ICS/LABA	2 (7.7)
ICS/LAMA/LABA	2 (7.7)

Data are shown as the mean ± SD, median (interquartile range), or number (%). LAMA, long-acting muscarinic antagonists; LABA, long-acting beta-agonists; ICS, inhaled corticosteroids.

**Table 3 ijms-25-00066-t003:** Enrichment analysis of transcriptomic data (COPD vs. healthy controls).

**A. Gene Ontology (biological process): Relevant terms were excerpted**		
**Term (Gene Ontology: biological process) with upregulated genes**	** *p* ** **-value**	**DEGs**
Regulation of chromatin binding (GO:0035561)	0.00074	*CDT1*, *DDX11*
Positive regulation of ERBB signaling pathway (GO:1901186)	0.00443	*NEU3*, *GPER1*
Acute inflammatory response (GO:0002526)	0.00518	*EHHADH*, *ELANE*
Negative regulation of wound healing (GO:0061045)	0.00600	*CD109*, *TFPI*
Positive regulation of epidermal growth factor receptor signaling (GO:0042058)	0.00827	*NEU3*, *GPER1*
Positive regulation of cell development (GO:0010720)	0.00926	*MME*, *GPER1*
Interstrand cross-link repair (GO:0036297)	0.00978	*DCLRE1A*, *NEIL3*
**Term (Gene Ontology: biological process) with downregulated genes**	** *p* ** **-value**	**DEGs**
Ameboidal-type cell migration (GO:0001667)	0.00059	*PKN3*, *AMOT*
Epithelial cell migration (GO:0010631)	0.00310	*PKN3*, *LOXL2*
Negative regulation of blood vessel morphogenesis (GO:2000181)	0.00847	*COL4A3*, *AMOT*
Positive regulation of Wnt signaling pathway, planar cell (GO:0060071)	0.00897	*NKD1*
Negative regulation of morphogenesis of an epithelium (GO:1905331)	0.00897	*NKD1*
**B. KEGG pathway: Relevant terms were excerpted**		
**Term (KEGG pathway) with upregulated genes**	** *p* ** **-value**	**DEGs**
Hematopoietic cell lineage	3.50 × 10^−6^	*HLA-DRB5*, *CSF1*, *MME*, *IL1R1*, *IL1R2*, *CD1A*
Protein digestion and absorption	0.00905	*CPA3*, *MME*, *COL24A1*
Complement and coagulation cascades	0.04867	*PROS1*, *TFPI*
NF-kappa B signaling pathway	0.06947	*IL1R1*, *BLNK*
Th17 cell differentiation	0.07298	*HLA-DRB5*, *IL1R1*
**Term (KEGG pathway) with downregulated genes**	** *p* ** **-value**	**DEGs**
Hippo signaling pathway	0.03470	*NKD1*, *AMOT*
Focal adhesion	0.05065	*MYLPF*, *COL4A3*
MAPK signaling pathway	0.09795	*PLA2G4C*, *CACNA2D2*

DEGs, differentially expressed genes.

**Table 4 ijms-25-00066-t004:** Characteristics of the patients with emphysematous and non-emphysematous COPD.

	Emphysematous COPD	Non-Emphysematous COPD	*p*-Value
Number of participants	14	8	N/A
Sex (male/female)	13/1	6/2	NS
Age (years)	69.3 ± 1.0	67.8 ± 0.6	NS
BMI (kg/m^2^)	22.7 ± 8.0	23.1 ± 6.2	NS
Current smokers, N (%)	4 (28.6)	2 (25.0)	NS
FEV_1_/FVC (%)	42.0 ± 2.8	61.2 ± 2.3	*p* = 0.0009
%FEV_1_ (%)	50.7 ± 1.7	76.7 ± 1.0	*p* = 0.0003
B.I.	1191.9 ± 71.5	943.6 ± 40.8	NS
Eosinophils (/μL)	293 ± 179.7	251.3 ± 245.4	NS
Eosinophils (%)	3.8 ± 2.4	3.8 ± 3.4	NS
Serum IgE (IU/mL)	197.0 ± 342.0 (*n* = 7)	188.9 ± 205.8 (*n* = 12)	NS
FeNO (ppb)	16.5 ± 5.5 (*n* = 6)	26.4 ± 32.4 (*n* = 10)	NS

Data are shown as the mean ± SD, median (interquartile range), or number (%). BMI, body mass index; B.I., brinkman index. N/A, not available. NS, not significant.

**Table 5 ijms-25-00066-t005:** Enrichment analysis of transcriptomic data (non-emphysematous vs. emphysematous COPD).

**A. Gene Ontology (biological process): relevant terms were excerpted**		
**Terms (Gene Ontology: biological process) with upregulated genes**	** *p* ** **-value**	**DEGs**
Positive regulation of T-helper 2 cell cytokine production (GO:2000553)	0.00075	*XCL1*, *PRKCZ*
Regulation of T-helper 2 Cell cytokine production (GO:2000551)	0.00158	*XCL1*, *PRKCZ*
Positive regulation of Type 2 immune Response (GO:0002830)	0.00271	*XCL1*, *PRKCZ*
Keratan sulfate biosynthetic process (GO:0018146)	0.00362	*B3GNT7*, *ST3GAL3*
Keratan sulfate metabolic process (GO:0042339)	0.00465	*B3GNT7*, *ST3GAL3*
Cellular response to inorganic substance (GO:0071241)	0.00521	*DDI2*, *AQP1*
Regulation of T cell cytokine production (GO:0002724)	0.00521	*XCL1*, *TRPM4*
Positive regulation of T cell cytokine production (GO:0002726)	0.00707	*XCL1*, *PRKCZ*
Regulation of angiogenesis (GO:0045765)	0.00821	*SERPINF1*, *ISM1*, *HSPG2*, *AQP1*, *PAK4*
Regulation of T cell migration (GO:2000404)	0.00845	*CD200R1*, *TMEM102*
Positive regulation of T cell migration (GO:2000406)	0.00994	*TMEM102*, *XCL1*
**Terms (Gene Ontology: biological process) with downregulated genes**	** *p* ** **-value**	**DEGs**
Regulation of vasoconstriction (GO:0019229)	0.00292	*KCNMB4*, *ADM*
Peptidyl-proline modification (GO:0018208)	0.00316	*P3H2*, *FKBP7*
**B. KEGG pathway: relevant terms were excerpted**		
**Terms (KEGG pathway) with upregulated genes**	** *p* ** **-value**	**DEGs**
ECM-receptor interaction	0.01633	*ITGA3*, *COL6A1*, *HSPG2*
Focal adhesion	0.03390	*PDGFRB*, *ITGA3*, *COL6A1*, *PAK4*
Hedgehog signaling pathway	0.04523	*GPR161*, *EFCAB7*
**Terms (KEGG pathway) with downregulated genes**	** *p* ** **-value**	**DEGs**
Vascular smooth muscle contraction	0.00817	*KCNMB4*, *ADM*, *MYLK4*
cGMP-PKG signaling pathway	0.09475	*KCNMB4*, *MYLK4*

## Data Availability

The datasets presented in this study can be found in online repositories. The names of the repository/repositories and accession number(s) can be found below: https://www.ncbi.nlm.nih.gov/, GSE248493.

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
