# Peer review of "Different Transcriptome Features of Peripheral Blood Mononuclear Cells in Non-Emphysematous Chronic Obstructive Pulmonary Disease"

_ijms, 2023, doi:10.3390/ijms25010066_

Round 1

Reviewer 1 Report

Comments and Suggestions for Authors

General

The manuscript describes the differences between gene expression profiles in peripheral blood mononuclear cells of emphysematous COPD non-emphysematous COPD. The authors used high-throughput mRNA sequencing technology to analyze alterations in transcripts abundance levels. The identified molecular differences were associated with multiple biological processes. In conclusion, the new findings are important for the etiology and pathobiology of COPD.

The study still lacks transparency in presentation and rigor in design. In the present form, the data are limited and descriptions of the results are too preliminary to be published.

Major concerns:

  1. Statistical analysis of the data

It is not clear if the obtained statistical results were adjusted for multiple comparisons by using an appropriate method for correction. Without this type of correction, there is a significant possibility that the presented results were obtained by chance. Both, corrected (using one of the available methods e.g. Bonferroni, FDR, or bootstrap) and nominal p values should be presented in the supplementary results. Moreover, the experiment was designed to compare three experimental groups and should be analyzed using ANOVA. The selection of statistical methods to analyze the data was not appropriately justified.

  1. Data presentation.

The figure presenting results of gene expression profiling are raw and obscure. For example, the heatmaps do not present gene names. Figure descriptions are too vague.

  1. Data availability

There is no information about the data availability in the GEO/SRA database. The ID number for this dataset should be provided in the manuscript.

  1. Novelty.

The manuscript is still descriptive. The novel aspects of this work have to be better exposed and justified. The description of the results is confusing (abstract: “could be associated”, “could augment”, “may augment”, “may contribute”, “could explain”, “seem to”). It is difficult to understand if anything new was really found in this study.

  1. Clinical implications

The results may suggest the possibility of using molecular profiling to stratify COPD patients. It is recommended to implement this study with the analysis of the predictive/differentiating potential of the identified gene expression signatures.

Comments on the Quality of English Language

No comments

Reviewer 2 Report

Comments and Suggestions for Authors

The abstract needs to be clearer and needs rewriting. It is unclear whether the findings are being expressed or it is a prelude to the findings. Suggest describe the differences between emphysematous and non-emphysematous COPD and then describe its relevance. Bringing other phenotypes like airway dominant and obstructive disordered fixed COPD is confusing and does not give clarity. Clearly state the findings in emphysematous and non-emphysematous COPD and what it means.

The baseline data of COPD emphysematous vs non- emphysematous shows a near normal FEV1/FVC%. This suggests that there was no obstruction. If so how do we say they had COPD?. Further severity should also be compared else the comparisons may not be valid as the two groups may be in different stages of the evolution of the disease.

In the discussion ACOS is mentioned as having less obstructive elements. This contradicts the definition/criteria that the FEV1/FVC ratio. Should be <70%. ( Ref:FED Pract 2015 Sep; 32 (suppl 10):19S-23S)

Material & methods -GINA guidelines were used. If so ratio of FEV1/FVC cannot be as high as 75%.

72.’used RNA’ should read ‘using RNA’

Comments on the Quality of English Language

Please review to make it more readable particularly the abstract.

Round 2

Reviewer 1 Report

Comments and Suggestions for Authors

The authors responded to the comments. The manuscript has been largely improved. I recommend publishing the paper.